# Brief communication: The anomalous winter 2019 sea-ice conditions in McMurdo Sound, Antarctica

Greg H. Leonard[1], Kate E. Turner[2, 3], Maren E. Richter[2], Maddy S. Whittaker[2], and Inga J. Smith[2]

[1]National School of Surveying, University of Otago, Dunedin, New Zealand
[2]Department of Physics, University of Otago, Dunedin, New Zealand
[3]National Institute of Water and Atmospheric Research, Wellington, New Zealand

**Correspondence:** Greg Leonard (greg.leonard@otago.ac.nz)

**Abstract.** McMurdo Sound sea ice can generally be partitioned into two regimes: (1) a stable fast-ice cover, forming south of approximately 77.6° S around March / April, then breaking out the following January / February; and, (2) a more dynamic region north of 77.6° S that the McMurdo Sound and Ross Sea polynyas regularly impact. In 2019, a stable fast-ice cover formed unusually late due to repeated break-out events. We analyse the 2019 sea-ice conditions and relate them to a Modified Storm Index (MSI), a proxy for southerly wind events. We find there is a strong correlation between the timing of break-out events and several unusually large MSI events.

## 1 Introduction

The sea-ice cover in McMurdo Sound can generally be partitioned into two regimes: (1) a stable fast-ice cover occupying the southeastern and western parts of the sound, south of a latitude of approximately 77.6° S; and, (2) a more dynamic region in the central part of the sound. Regime (1) is primarily made up of first-year sea ice that forms around late March / early April and typically breaks out in January / February of the following year (Kim et al., 2018) whereas Regime (2) is impacted by the McMurdo Sound Polynya (MSP) and Ross Sea Polynya (RSP) (Brett et al., 2020). The boundary between the regimes is influenced by such factors as ice-shelf–ocean interactions, ocean circulation (e.g., Hughes et al., 2014; Robinson et al., 2014) and the location of grounded icebergs (e.g., Brunt et al., 2006; Robinson and Williams, 2012) and can vary on annual timescales.

McMurdo Sound logistical and scientific sea-ice operations depend on the formation of a stable fast-ice cover over the winter months that persists through to late December / early January. In 2019, the formation of the stable fast-ice cover was significantly delayed, impacting operations for both the New Zealand and the United States of America Antarctic programmes. Impacts included: the non-establishment of a sea-ice route to Marble Point (a cache for helicopter fuel and other supplies); a reduction in the number of sea-ice scientific field sites; and a two-month delay in the deployment of the University of Otago sea-ice mass balance station.

We are not aware of any studies directly investigating the causes of delayed formation of stable fast-ice covers in McMurdo Sound. Kim et al. (2018) found that in years with higher mean annual wind speeds, the fast ice generally retreated earlier in the season, however, that study did not look at the impact of individual events on break-out and retreat. At the event level, investigations by Banwell et al. (2017) into causes of the calving of the McMurdo Ice Shelf in 2016 suggested that strong (>10 m s$^{-1}$) winds from the south and west contributed to the large fast-ice break-out event that preceded the calving event. Brunt et al. (2006) investigated sea-ice break-out events in the southwest Ross Sea between 1996 and 2005 using satellite imagery and automatic weather station data. They found that break-out events were correlated with a dimensionless "storm index", defined as the product of low-pressure anomalies and anomalous temperature (lower temperatures in summer and higher temperatures in winter).

In this brief communication we investigate the linkages between more frequent opening of the MSP due to increased frequency of intense winter storms, and the delay in the formation of a stable fast-ice cover in 2019. We quantify winter storm intensity by introducing a Modified Storm Index (MSI) that is based on the approach of Brunt et al. (2006), but here is applied to individual events as opposed to seasonal trends. The 2019 McMurdo Sound sea-ice cover properties are derived from a combination of manual assessment of Sentinel-1 Synthetic Aperture Radar (SAR) and MODIS-derived ice-surface temperatures and analysing sea-ice concentrations generated by the ARTIST Sea Ice (ASI) algorithm (Spreen et al., 2008).

## 2   McMurdo Sound sea-ice characteristics

The edge of the stable fast-ice cover typically extends west-to-east from a band of fast ice ~20 km wide along the Victoria Land Coast to Cape Royds, Ross Island (see Figure 1a). To investigate when this sea-ice cover forms, we analysed the Fraser et al. (2020) circum-Antarctic land-fast sea-ice distribution data set derived from cloud-free, 15-day composites of satellite visible–thermal infrared imagery. This data set covers March 2000 to February 2018 and includes the time when the sea ice in McMurdo Sound was strongly impacted by the presence of very large, tabular icebergs in the mouth of the sound (e.g., B-15A from approximately January 2001 to November 2004 (Brunt et al., 2006) and C-19 from approximately September 2002 to May 2003 (Arrigo and van Dijken, 2003)).

From 2001 to 2005, the sea ice (Brunt et al., 2006) and ocean circulation (Robinson and Williams, 2012) in McMurdo Sound were directly influenced by these icebergs, leading to the presence of multi-year ice and anomalously large sea-ice extents that influenced the studies of Brunt et al. (2006) and Kim et al. (2018). The effects on the fast ice in McMurdo Sound were felt for several years after the icebergs exited the region, as evidenced by the fact that a multi-year fast-ice cover remained in place in the southern part of the sound until February 2011. Therefore, the fast-ice extents for these years (2001 – 2011) have been excluded from this analysis.

In years that were not iceberg-affected (2000; 2012 – 2017), the fast-ice cover in the sound generally reached a minimum in mid-March where it typically receded into a few pockets along the Victoria Land Coast in the west of the sound, around the Erebus Glacier Tongue along the west coast of Ross Island and into a wedge-shaped area between the tip of the Hut Point Peninsula on Ross Island and the McMurdo Ice Shelf. The fast-ice cover south of 77.6° S typically re-formed sometime

between mid-March and mid-April, with the exception of 2012, where it did not form until around the end of June. This is contrasted with 2019 when a stable fast-ice cover did not form until late July.

## 2.1   2019 sea-ice conditions

An analysis of radar (Sentinel-1 SAR) imagery and MODIS 1-km resolution Ice Surface Temperature (IST) (Hall and Riggs, 2015a, b) during April – July 2019 revealed an unusually large number of MSP events (a "polynya event" being defined here as a polynya opening that impacts on the previously established fast-ice cover). The frequency and intensity of these events resulted in the eroding of the fast-ice cover all the way back to the edge of the ice shelf, ~30 km farther south than 77.6° S. The SAR imagery used in this study were a combination of Extra Wide (EW) medium resolution imagery (40 m pixel size) and Interferometric Wide (IW) high resolution mode imagery (10 m pixel size resampled to 40 m pixel size for this study). From 1 April to 1 September, eight large MSP events were observed in SAR images, three of which are shown in Figure 1b – d (21 May, 25 June and 8 July). Prior to the events, MODIS IST pixels in the east of the sound typically show increased temperature values, although we are unable to determine whether these higher ISTs resulted from increased ice surface temperatures or sea ice within these pixels being dynamically broken-up and advected northward due to wind forcing (hence exposing underlying warm ocean water), or a combination of the two. Due to this, and the well-documented issues with cloud masking in the MODIS IST product (e.g., Mäkynen and Karvonen, 2017), we used MODIS IST as a secondary source to corroborate SAR observations rather than the primary source to define fast-ice break-out events. However, by qualitatively comparing spatial patterns of elevated IST pixels with coincident (when available) SAR imagery during break-out events, we observed that elevated IST pixels typically occur in areas in the eastern sound where the MSP was active, suggesting that increased ISTs are at least partially a result of polynya activity.

Manually-identified fast-ice break-out events were contextualised with sea-ice concentrations derived from the ARTIST Sea Ice (ASI) algorithm (Spreen et al., 2008). Figures 1e and f show sea-ice concentrations for the late-June and mid-July event, respectively. The ARTIST land mask extends up to ~15 km out into the southern sound, hence sea-ice concentrations cannot be determined in this area from ASI data. Due to the challenges of discriminating between thin ice and fast ice in passive microwave satellite brightness temperatures, (e.g., Tamura et al., 2007, 2008; Nihashi and Ohshima, 2015), an attempt was not made to quantify polynya area from the ARTIST derived sea-ice concentrations. Instead, a daily sea-ice fraction metric (see Section 4) is introduced to characterise regional changes in sea-ice extent within McMurdo Sound. In comparing Figure 1c to Figure 1e it can be seen that the late-June event broke up the fast-ice cover all the way back to the edge of the ice shelf, but this has not been fully captured in the ARTIST sea-ice concentration due to its land mask. This delayed the impact of the late-June event on the ARTIST sea-ice concentrations by a couple of days, which will be discussed further in Section 4.

## 3   Characterisation of winter storms

Strong surface-wind events in McMurdo Sound can develop when southerly katabatic and barrier winds that flow off the Ross Ice Shelf (e.g., Coggins et al., 2013; Parish et al., 2006) interact with synoptic-scale low pressure systems forming to the east of

Ross Island. In winter, these winds are typically associated with low air pressure and warm near-surface air temperatures (e.g., Coggins et al., 2013; Chenoli et al., 2013). The relative warmth of the surface winds results from high wind speeds (above 4 – 6 m s$^{-1}$) increasing the vertical mixing between the relatively cold surface inversion layer over the ice shelf and the warmer overlying atmosphere (Cassano et al., 2016) and warm, marine air being incorporated from the synoptic-scale lows. Dale et al. (2017) found that southerly wind events in the Ross Sea are correlated with low sea-ice concentration and the opening of the Ross Sea Polynya (RSP), which they attributed to strong winds causing northward advection of sea ice. They also observed that a rapid decrease in sea-ice concentration during a strong wind event was followed by a more gradual recovery. Ebner et al. (2013) undertook mesoscale atmospheric model simulations coupled with a sea-ice–ocean model to investigate the formation of polynyas in the coastal region off Coats Land, which is an area strongly affected by katabatic winds. They identified linkages between a pressure gradient force composed of a katabatic and a synoptic component, offshore wind regimes and polynya area. Here we examine the relationship between strong surface-wind events in McMurdo Sound and the opening of the MSP in the winter of 2019.

The available observational wind data for Scott Base (Figure 2) show that the months preceding the formation of a stable fast-ice cover in McMurdo Sound in 2019 were characterised by particularly strong southerly winds. Columns 1 and 2 of Figure 2 indicate that southerly winds for these months in 2019 were more frequent, and in the case of the June and July months, stronger, than the average of previous years (1997 – 2018). Columns 3 and 4 further highlight particularly strong (up to 30 m s$^{-1}$) winds during June and July, well above their climatological 90th percentiles, (10.3 and 10.0 m s$^{-1}$, respectively). These strong wind events can be seen in Figure 3c and d, corresponding with the timing of the fast-ice break-out events as identified through satellite imagery.

To investigate this further, following the approach of Brunt et al. (2006), we used hourly air temperature and mean sea-level pressure (MSLP) data from the Scott Base weather station to construct a dimensionless Modified Storm Index (MSI). The Scott Base weather station is located immediately adjacent to Scott Base at 77.85° S, 166.76° E, approximately 150 m from the coast and on the order of 30 km east of the centre of the sound. The elevation of the station is 20 m AMSL with the sensors deployed at a height of 6–7 m above the ground. The ground surface is is seasonally snow-free scoria and volcanic rock. Mean hourly climatologies of temperature and MSLP were constructed by applying a three-day smoothing window to the mean of hourly observations from 2002 – 2018. As the period of interest spans only the winter months, we define the MSI as the product of normalised positive air temperature and negative MSLP anomalies, as calculated in relation to their climatological means. The MSI was then smoothed over a 12-hour window. The index, and the corresponding air temperature and MSLP data, are shown in Figure 3.

As defined by this index, in the period 1 April to 1 September, three large storm events (MSI > 0.20) leading to MSP openings occurred in mid-June, late-June and mid-July of 2019, again coincident with times when large break-out events were identified in the Sentinel-1 SAR imagery. Smaller events can also be seen in the preceding months, largely concurrent with Sentinel-1 implied break-out events. After the final freeze-in occurred in late July, defined as the earliest time after which no further significant break-out events were observed in the SAR imagery, the storm events were not not strong enough to lead to significant break-out of the fast ice.

## 4 Impact of mid-winter southerly wind events on the 2019 fast-ice cover

To quantify the effect of these storm events on the 2019 sea-ice cover, we first computed a daily sea-ice fraction (which we assigned to mid-day) for McMurdo Sound for 1 April to 1 September over the entire ARTIST record (2013 – 2019). The sea-ice fraction was calculated by summing the number of pixels within the bounds of the red box shown in Figure 1a where the sea-ice concentration was 15 % or greater (a typical threshold used to determine sea-ice extent) and dividing by the total number of pixels (n = 553), excluding land masked pixels. This method was chosen in preference to computing an average sea-ice concentration as it is better suited to quantifying changes in sea-ice extent, which we equate generally to changes in the fast-ice cover in the sound.

The sea-ice fractions for the years 2013 – 2019 are shown in Figure 3a, with the 2019 data indicated by a thick black line. It can be clearly seen that the timing of low (< 0.80) sea-ice fraction events correspond with identified sea-ice break-out events, even though the sea-ice fraction is likely underestimating the amount of open water due to the protrusion of the ARTIST land mask into the southern reaches of the sound. An example of this is the late-June event (reference Figure 1c) where the sea-ice fraction decrease lags the MSI and the Sentinel-1 SAR data identified break-out event by a couple of days. This break-out event was characterised by the MSP first opening just off the edge of the ice shelf (which is masked out in the ARTIST sea-ice concentrations) before expanding northward, whereas in other events the polynya tends to be centred farther to the north. The 2019 record stands apart from other years, both in terms of the number of low sea-ice fraction events overall and it being the only year where sea-ice fractions in June and July decreased below 0.75.

## 5 Discussion

The strength of a sea-ice cover is impacted by factors such as its thickness, internal temperature, salinity and porosity (Timco and Weeks, 2010), and individual fast-ice break-out events can be triggered by strong surface winds and ocean currents, upwelling, waves or tides. Although we cannot unequivocally rule out other break-out mechanisms than wind forcing for any particular break-out event due to a lack of data, none offer a particularly compelling alternative hypothesis. Kim et al. (2018) did not find a significant relationship between sea-ice temperature and break-out in their McMurdo Sound study, and we did not find any correlation between tidal state and the 2019 break-out events in this study, based on our analysis of 5-minute sea-level height data from the Land Information New Zealand tide gauge at Cape Roberts (77.0 ° S, 163.2 ° E) in the northwestern sound. Furthermore, the entire water column in McMurdo Sound during winter is conditioned by supercooled water flowing out from the McMurdo Ice Shelf cavity (Leonard et al., 2006), resulting in it being nearly isothermal and very close to its freezing point (Lewis and Perkin, 1985; Mahoney et al., 2011). This suggests that any water upwelled from the opening of the MSP would not cause melting of the fast-ice cover. Finally, we do not anticipate wave action playing a significant role in breaking up the fast-ice cover due to the absence of upstream fetch associated with the southerly winds that are typical of these 2019 break-out events.

Our analysis implies that the persistence of the McMurdo Sound fast-ice cover south of 77.6° S is vulnerable to storm events from late spring to mid-winter. The fast-ice cover tends to respond to storm events in April and May, presumably because the

ice cover is still relatively thin and warm and lacks the strength to resist the drag force from the southerly winds. The MSI climatology suggests the frequency of intense southerly storms tends to decrease in the months of June and July. This typically allows the fast-ice cover to thicken and gain strength during these months, which means that it is sufficiently strong to remain intact when the southerly winds increase, on average, later in the year. However, in 2019, the frequency of intense storms in June and July was such that the fast-ice cover was too weak to withstand the southerly winds and was broken up and advected northwards on three separate occasions, activating the McMurdo Sound Polynya. Immediately following the mid-July event, the MSI and wind data indicate a more quiescent weather pattern, allowing the fast-ice cover to reform and gain strength.

The MSI and wind speed increased again in the second half of August, however, the fast-ice cover south of 77.6° S was now able to resist the southerly wind stress, staying fixed in place for the remainder of the growth season. Brett et al. (2020) explored reasons for an anomalously extensive fast-ice cover in McMurdo Sound in 2016 and related relatively low activity of the RSP and MSP over winter with calmer conditions that led to largely undisturbed sea-ice growth in 2016. Here, we have observed the opposite scenario where a more active MSP, resulting from an increase in the frequency of intense winter storms, has resulted in an anomalously dynamic fast-ice cover. Both studies clearly illustrate the impact winter storms have on the fast-ice cover in McMurdo Sound.

## 6   Conclusions

Here we have investigated the impact of winter storms on the stability of an Antarctic fast-ice cover. Our key finding is that an increase in the frequency of intense winter storms in 2019 resulted in a delayed formation of a stable fast-ice cover in McMurdo Sound. This study offers new insights into the mechanisms behind individual break-out events and is one of a few case studies that investigate the stability of a fast-ice cover, an area that is in need of future research to improve the parameterisation of fast-ice processes in large-scale sea-ice models. Furthermore, improved understanding the drivers of fast-ice break-out is critical to informing safe on-ice operations by national Antarctic programmes. McMurdo Sound is a well-suited location for these types of studies as it has extensive and accessible fast ice and is a location where important US and New Zealand Antarctic programme field operations are undertaken; all of which are vulnerable to changes in the fast-ice cover. It is also a unique environment in that it is a relatively deep embayment that experiences cold water outflow from an adjacent ice shelf cavity that leads to the formation of a sub-ice platelet layer, making it an ideal location to investigate ice-shelf–ocean–fast-ice feedbacks along the Antarctic coastal margin. Future work will identify how the MSI and fast-ice extent in McMurdo Sound has varied throughout the observational record and how it might change in the future.

*Data availability.* Scott Base weather station data are available from https://cliflo.niwa.co.nz/. Copernicus Sentinel-1 data, retrieved from the Alaska SAR Facility DAAC at https://search.asf.alaska.edu/, were processed by the European Space Agency. MODIS images were accessed through https://urs.earthdata.nasa.gov. ARTIST sea-ice concentration data accessed from the University of Bremen data archive at https://seaice.uni-bremen.de/data/. Circum-Antarctic landfast sea-ice extent data accessed from the Australian Antarctic Data Centre at

https://data.aad.gov.au/metadata/records/AAS_4116_Fraser_fastice_circumantarctic. Cape Roberts tide gauge data can be accessed from Land Information New Zealand at https://www.linz.govt.nz/sea/tides/sea-level-data/sea-level-data-downloads.

*Author contributions.* GHL conceived this study and analysed the satellite images/products. KET analysed the climatological data in reference to the break-out events and designed the MSI. MER analysed the Scott Base weather station climatology. MSW identified break out events in 2019 from SAR and MODIS images. GHL, KET, MER and IJS discussed the results and prepared the manuscript. All authors contributed to the final review of the manuscript.

*Competing interests.* There are no competing interests.

*Acknowledgements.* This paper is an output from the project "Supercooling measurements under ice shelves" supported by the Marsden Fund Council from Government funding, administered by the Royal Society of New Zealand (Marsden Fund contract number MFP-UOO1825). MSW was supported on a summer scholarship from that project, and IJS and GHL's time was supported by that project. MER is supported by a University of Otago Doctoral Scholarship and an Antarctica New Zealand Sir Robin Irvine Post-Graduate Scholarship. The authors are very grateful to Alex Fraser, an anonymous reviewer and the editor for their constructive comments. The authors further appreciate a discussion with David Bromwich regarding the wind regime in McMurdo Sound.

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

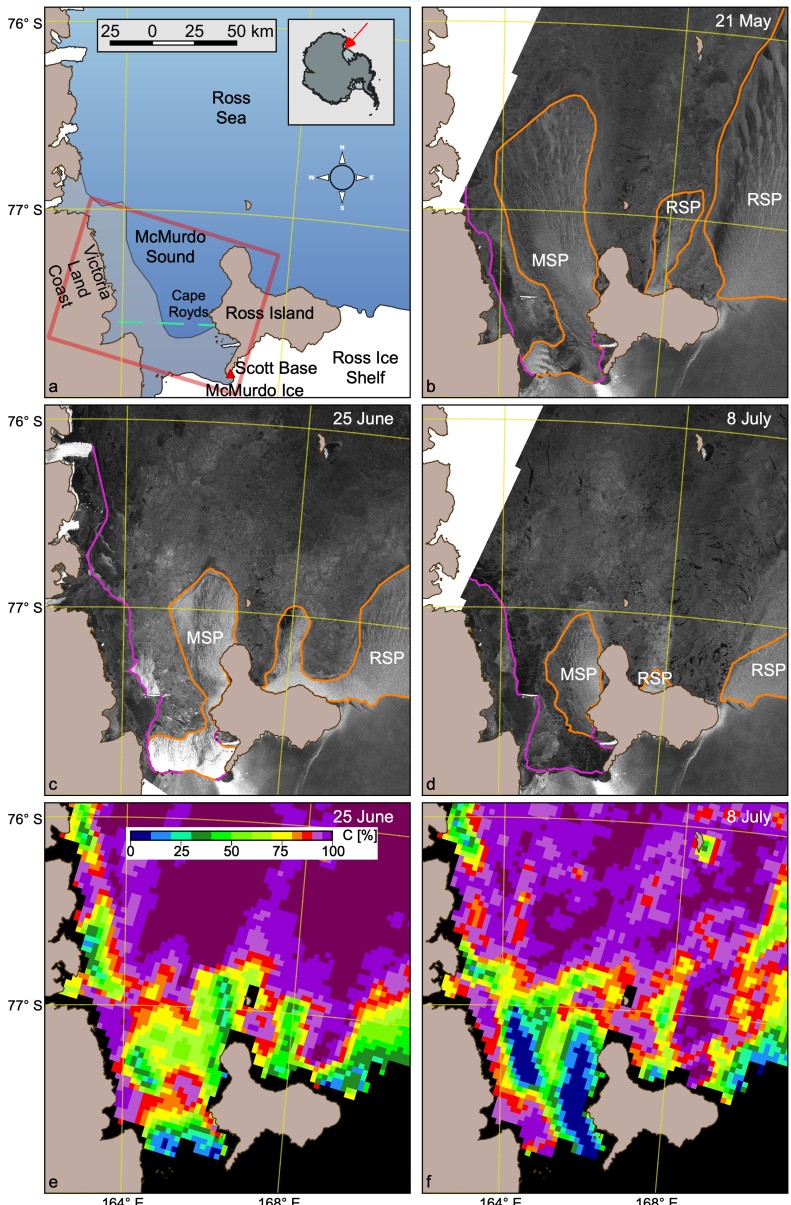

**Figure 1.** (a) McMurdo Sound study region, inset shows the location in the Western Ross Sea, Antarctica. The red box indicates the area for which the sea-ice fractions were calculated. The grey shaded area indicates fast-ice extent from the middle of June 2014 from Fraser et al. (2020) and the green dashed line shows the 77.6 ° S parallel. (b) Sentinel-1 SAR image from 21 May 2019, (c) 25 June 2019 and (d) 8 July 2019, with the hatched areas with the orange border indicating the active area of the McMurdo Sound Polynya (MSP) and Ross Sea Polynya (RSP) and the magenta line depicting the fast-ice edge. (e) Sea-ice concentrations from 25 June 2019 and (f) 8 July 2019 from the ARTIST Sea Ice (ASI) algorithm (Spreen et al., 2008).

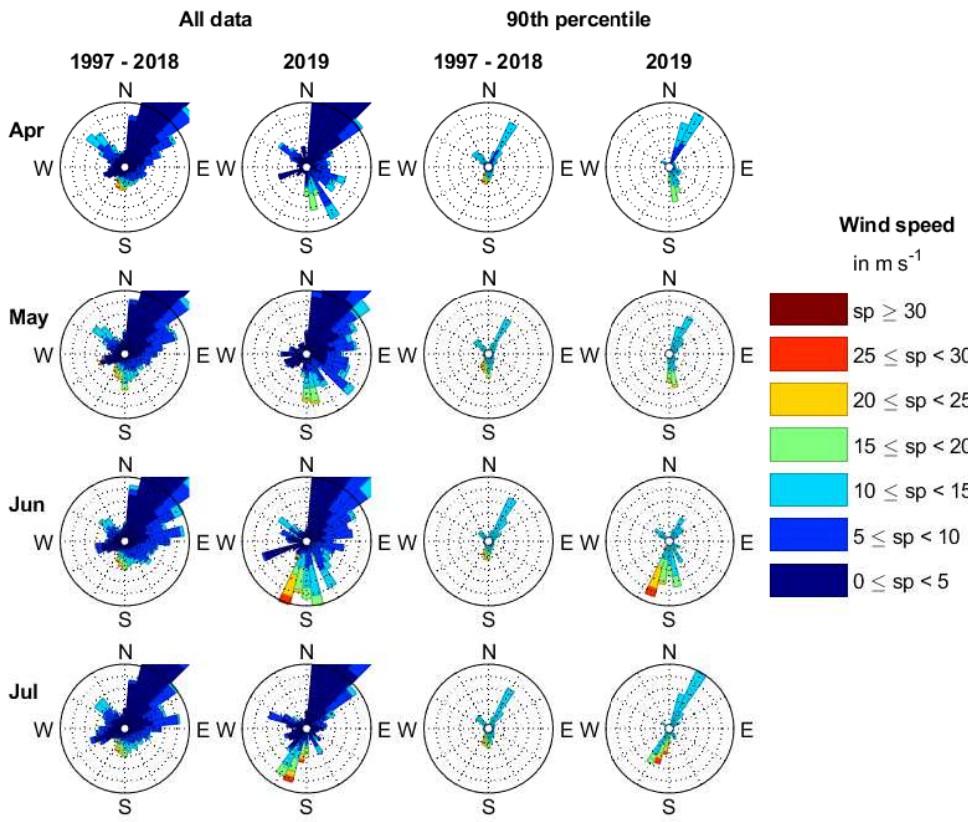

**Figure 2.** Wind roses constructed from Scott Base weather station data for the months of April – July. The first and second columns show the full data set for 1997 – 2018 and 2019, respectively, and the third and fourth columns show wind speeds equal to or greater than the monthly 90th percentile for 1997 – 2018 and 2019, respectively. The maximum frequency shown is 3 %, with each circle representing an increment of 0.6 %, and the wind direction defined as the bearing from which the wind originated. The bin size of each spoke is 10°. Note that in the first two columns the prevailing north-easterly winds extend to a maximum of  30 % (not shown) for all months.

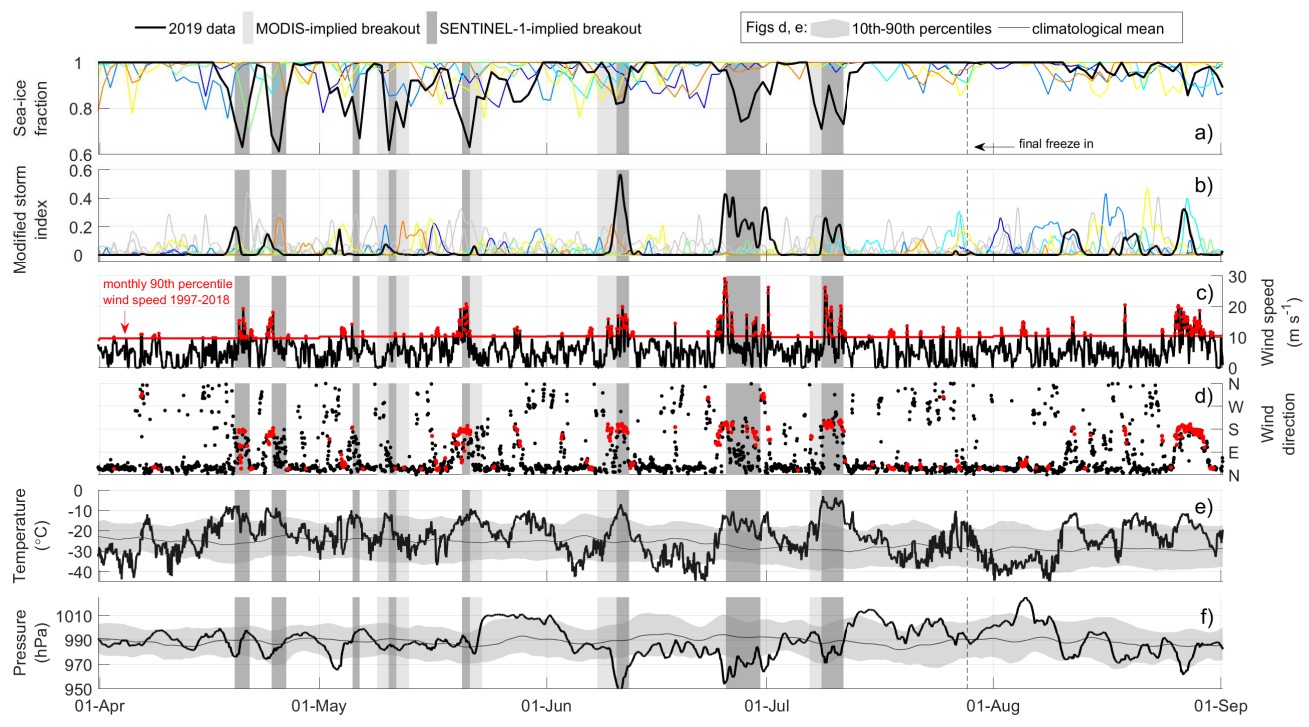

**Figure 3.** a) Sea-ice fractional cover within the red bounding box shown in Fig. 1.a for years 2013 – 2019; Black: 2019, coloured lines: 2013 – 2018. b) Modified Storm Index (MSI) constructed from temperature and mean sea-level pressure anomalies for years 2002 – 2019; Black and coloured lines as for a), grey: years for which sea-ice fraction data are not available (2002 – 2012). c) 2019 wind speed; Red line: the monthly 90[th] percentile wind speed (data from 1997 – 2018), Red markers: wind speeds greater than the climatological 90[th] percentile. d) 2019 wind direction; Red markers: wind speeds greater than the climatological 90[th] percentile. e–f) Surface air temperature and MSLP, respectively. The 10[th] – 90[th] percentiles and climatological mean include temperature and MSLP data from 1997 – 2018 and 2002 – 2018, respectively. Percentiles have been smoothed using a 3-day window. Vertical shading throughout all subplots identifies times where satellite products indicate fast-ice break-out events (see Section 2.1).