# Peer review of "Brief communication: The anomalous winter 2019 sea ice conditions in McMurdo Sound, Antarctica"

_The Cryosphere, 2020_

## Author Comment (AC1)

**Final Author Comments - responses to Reviewer 1 comments**

21 April 2021

**1  Responses to Reviewer 1 Comments**

We thank Reviewer 1 for their comments on our submission as a Brief Communication within criteria (c): "to disseminate information and data on topical events of significant scientific and/or social interest". We provide the following responses to the points that they have raised.

Reviewer 1 requested that alternative fast ice breakout mechanisms to wind-driven breakout be briefly discussed in the manuscript. Reviewer 2 has also requested more explanation on the fast ice breakout mechanism as well as consideration of other breakout mechanisms.

We agree with both reviewers that these would be valuable additions to the manuscript and will look to include an expanded discussion on breakout mechanisms in the revised manuscript that will include:

1.) Describing how offshore surface winds in McMurdo Sound activate the McMurdo Sound Polynya (MSP), which influences the fast ice cover in the sound. This discussion will reference the findings of Ebner et al. (2013) regarding surface wind forcing and polynya opening and draw on the findings of Dale et al. (2017) who found strong negative correlations between sea ice concentration (SIC) and AWS wind speed data within the Ross Sea Polynya (RSP), which they attributed to strong winds causing advection of sea ice in the region. Dale et al. (2017) also observed that a rapid decrease in SIC during a strong wind event was followed by a more gradual recovery in SIC. The MSP is proximal to the RSP and typically activates under the same strong offshore (southerly) wind conditions.

2.) A brief investigation of other potential breakout mechanisms, including thermodynamic, as referenced by Reviewer 1, and sea swell, as referenced by Reviewer 2, as a potential breakout mechanism. Regarding themo-dynamic drivers, we note that the entire water column in McMurdo Sound during winter is characterised by cold and very cold (supercooled) water outflow from underneath the McMurdo Ice Shelf (Leonard et al., 2006), and the water column is nearly isothermal and very close to its freezing point (Lewis and Perkin, 1985; Mahoney et al., 2011). During summer, surface waters flowing into the sound from the northeast exhibit higher temperatures (above the freezing point but mainly below 0 °C) (Robinson et al., 2014), but this is not the case during the months this study examines. Regarding sea swell, we agree that a study of sea swell effects on McMurdo Sound fast ice would be profitable. However, we note that there was no sea swell data available during the period of our study, as the frequency at which the sea level was recorded by the tide gauge was not sufficient to resolve waves. Thus, the available data in the region and the scope of the manuscript did not allow us to study the effect of sea swell on the McMurdo Sound fast ice in 2019 or the synoptic weather patterns associated with the AWS data. We agree with Reviewer 2 that due to the absence of fetch upstream of the fast ice in the direction of the wind, no significant swell can be formed prior to the polynya having opened. We note that previous studies on fast ice break up in Antarctica did find that the fast ice broke without a clear association to a wave event (Voermans et al., 2020).

Regarding the minor comments made by Reviewer 1, we generally accept the suggestions made will improve the manuscript and thank the reviewer for them. We make the following comment on some of the particular suggestions. Reviewer comments have been italicized for clarity. The numbers refer to the line numbers in the manuscript.

R1 comment:

*47: The "biased" in here implies that these studies didn't correctly account for the icescape change. Is this what you really mean - if so, for both studies?*

Our response:

We did not wish to imply that the studies did not correctly take into account the effect of the icebergs. Rather what we meant was that the time-period over which both of these studies investigated the sea ice in McMurdo Sound was influenced by the icebergs and that they thus only have limited ability to describe the normal, non-iceberg state.

R1 comment:

*58: Was this IW mode Sentinel-1 imagery? What resolution?*

Our response:

The three SAR images shown in Figure 1 are Extra Wide (EW) medium resolution mode with a pixel spacing of 40 m. The wider set of SAR imagery used to identify the presence of the McMurdo Sound Polynya also included some Interferometric Wide (IW) high resolution mode images with a native pixel spacing of 10 m that were re-sampled to 40 m pixel spacing for this study.

R1 comment:

*59: "MSP event" is a little ambiguous. Do you mean a large polynya size event?*

Our response:

Yes, we mean a large polynya size event, where the polynya impacts the previously established fast ice cover, breaking it up, rather than where a polynya is formed offshore of the fast ice edge.

R1 comment:

*Also here, I'm curious how an active polynya looks in ice surface temperature - presumably a warm temperature? Or is it masked because largely open water?*

Our response:

Yes, it is warm, but there are well known issues due to cloud masking with the MODIS Ice Surface Temperature (IST) products (MYD029 and MOD029) that make it difficult to resolve the area of the active polynya with MODIS IST. We did see warming in the thermal imagery generally correlating to polynya breakout, but it was not always clear. Hence, we used thermal imagery as a secondary source (to corroborate SAR observations) rather than the primary source to define fast ice breakout events.

R1 comment:

*117: Although a brief communication, the "big picture" could do with a little more expansion. E.g., this is one of few case studies on fast ice stability, an area where more research is needed, etc. It occurs to me that this region might be a good one for testing forthcoming fast ice tensile strength parameterisations in prognostic fast ice models (e.g., Lemieux et al., 2016, "Improving the simulation of landfast ice by combining tensile strength and a parameterization for grounded ridges"). Also, are there other regions you know of which have a similar fast ice regime (i.e., deep embayment and lack of grounded icebergs) to which the results of this study might be applicable?*

Our response:

We agree with that McMurdo Sound would make a fascinating site for testing fast ice break-out processes. Other areas where similar conditions are found would be Arctic fjords, though here the influence of warm water intrusions from below could potentially play an additional role. As far as we are aware, McMurdo Sound is unique in being a deep embayment experiencing strong cold water outflow from an adjacent ice shelf and not influenced by large ice tongues or icebergs. However, a study into wind regimes facilitating fast ice break out would be profitable for other fast ice locations both in the Arctic and Antarctic.

R1 comment:

*Fig 1: It would be helpful to please annotate the area of active polynya in each SAR image (manually is fine). Similarly for the fast ice edge.*

Our response:

We think this a great suggestion and will do this for the revised manuscript.

R1 comment:

*Fig 2: Does the truncation of the upper half of each wind rose remove any/much information? I'd quite like to see the whole thing (if there's detail in the northerly half) but happy to stick with the half roses if no wind from that half.*

Our response:

The northern half of the wind roses for both columns (all years, and 2019 only) are saturated with high-frequency, low-speed ($0 - 10$ m s$^{-1}$) NE winds, indicating that these are the prevailing surface wind conditions at the site. As the inclusion of these winds obscures the comparatively large increases in (discrete) high-speed surface wind events from the south (as the maximum frequency of the wind roses has to increase dramatically to include them), we believe that the inclusion of the northern half of the wind roses is counter-productive to the data narrative.

Below we have included two versions of the full wind roses to illustrate this point; one with a 3% maximum frequency (see Figure 1) and one with a 30% maximum frequency (see Figure 2). We will emphasise this point in the revised manuscript by adding the following sentence to the figure caption "The low-speed ($0 - 10$ m s$^{-1}$) prevailing winds from the NE are not shown, in order to highlight the comparative increase in strong but short-lived extreme winds from the south in 2019."

[Figure]

Figure 1: Full wind roses using the same scaling as per Figure 2 in the manuscript.

[Figure]

Figure 2: Full wind roses with an adjusted scaling that is required to show all of the NE winds.

R1 comment:

*Fig 3: A little unusual to not have a colour legend for the upper two plots, although I recognise that they're only shown to indicate the envelope of previous years (and the reader doesn't necessarily need to know which year is which).*

Our response:

We acknowledge it is unusual not to have a legend for all figure objects, however, we believe that the particular years in the data record that each timeseries relates to is not important, and a legend would complicate the figure. However, the ability to compare the same year (same colour) between sub-figures a) and b) is important, as it allows the reader to perceive that within the data record (2002 – 2019), 2019 contains the largest number of KWI events. This is an important aspect of the manuscript as the KWI index extends further back in time than the relatively shorter sea ice concentration dataset. So although we are not able to relate sea ice concentrations to KWI prior to 2013, we can show that the KWI signature in 2019 was significantly different from the other years in the record.

For this reason, we do not wish to apply a common colour for all years other than 2019 when sea ice fraction data is available (2013 – 2018), and propose to retain the current figure layout with individual colours and no legend.

**2   References**

Bogardus, R., Maio, C., Mason, O., Buzard, R., Mahoney, A. and de Wit, C.: Mid-Winter Breakout of Landfast Sea Ice and Major Storm Leads to Significant Ice Push Event Along Chuckchi Sea Coastline, Front. Earth Sci., https://doi.org/10.3389/feart.2020.00344, 2020.

Brunt, K. M., Sergienko, O., and MacAyeal, D. R.: Observations of unusual fast-ice conditions in the southwest Ross Sea, Antarctica: preliminary analysis of iceberg and storminess effects, Annals of Glaciology, 44, 183–187, https://doi.org/10.3189/172756406781811754, 2006.

Dale, E.: Interactions Between Changing Weather Patterns and the Antarctic Cryosphere in the Ross Sea Region, PhD Thesis, University of Canterbury, New Zealand, 2020.

Dale, E. R., McDonald, A. J., Coggins, J. H. J., and Rack, W.: Atmospheric forcing of sea ice anomalies in the Ross Sea polynya region, The Cryosphere, 11, 267–280, https://doi.org/10.5194/tc-11-267-2017, 2017.

Ebner, L., Heinemann, G., Haid, V. and Timmermann, R.: Katabatic winds and polynya dynamics at Coats Land, Antarctica, Antarctic Science, 26(3), 309-326, https://doi:10.1017/S0954102013000679, 2013.

Leonard, G H., Purdie, C. R., Langhorne, P. J., Haskell, T. G., Williams, M. M. J., Frew, R. D.: Observations of platelet ice growth and oceanographic conditions during the winter of 2003 in McMurdo Sound, Antarctica, J. Geophys. Res., 111, C04012, doi:10.1029/2005JC002952, 2006.

Lewis, E. L. and Perkin, R. G.: The Winter Oceanography of McMurdo Sound, Antarctica. In Oceanology of the Antarctic Continental Shelf, S.S. Jacobs (Ed.), https://doi.org/10.1029/AR043p0145, 1985.

Mahoney, A. R, Gough, A. J., Langhorne, P. J., Robinson, N. J., Stevens, C. L, Williams, M. M. J., and Haskell, T. G.: The seasonal appearance of ice shelf water in coastal Antactica and its effect on sea ice growth, J. Geophys. Res., 116, C11032, doi:10.1029/2011JC007060, 2011.

Robinson, N. J., Williams, M. J. M., Stevens, C. L., Langhorne, P. J., and Haskell, T. G.: Evolution of a supercooled IceShelf Water plume with an actively growing subice platelet matrix, Journal of Geophysical Research: Oceans, 119, 3425–3446, https://doi.org/10.1002/2013JC009399, 2014.

Voermans, J., Rabault, J., Filchuk, K., Ryzhov, I., Heil, P., Marchenko, A., Collins, C., Dabboor, M., Sutherland, G., and Babanin, A.: Experimental evidence for a universal threshold characterizing wave-induced sea ice break-up, The Cryosphere Discussions, pp. 1–24, https://doi.org/10.5194/tc-2020-201, 2020.

---

## Author Comment (AC2)

**Final Author Comments - responses to Reviewer 2 comments**

21 April 2021

**1 Responses to Reviewer 2 Comments**

We thank Reviewer 2 for their comments on our submission as a Brief Communication within criteria (c): "to disseminate information and data on topical events of significant scientific and/or social interest". We provide the following responses to the points they have raised.

Reviewer 2 commented generally that the breaking mechanism of fast ice was not discussed adequately in the manuscript. As noted in our Final Author Comments on Reviewer 1's comments, we agree with both reviewers that further discussion of fast ice break out mechanisms would be valuable additions to the manuscript and we will look to include an expanded discussion on breakout mechanisms in the revised manuscript. We have responded to the general comments of both reviewers in our Final Author Comments on Reviewer 1's comments, and included our responses here for completeness.

We agree with both reviewers that further discussion of fast ice break out mechanisms would be valuable additions to the manuscript and will look to include an expanded discussion on breakout mechanisms in the revised manuscript that will include:

1.) Describing how offshore surface winds in McMurdo Sound activate the McMurdo Sound Polynya (MSP), which influences the fast ice cover in the sound. This discussion will reference the findings of Ebner et al. (2013) regarding surface wind forcing and polynya opening and draw on the findings of Dale et al. (2017) who found strong negative correlations between sea ice concentration (SIC) and AWS wind speed data within the Ross Sea Polynya (RSP), which they attributed to strong winds causing advection of sea ice in the region. Dale et al. (2017) also observed that a rapid decrease in SIC during a strong wind event was followed by a more gradual recovery in SIC. The MSP is proximal to the RSP and typically activates under the same strong offshore (southerly) wind conditions.

2.) A brief investigation of other potential breakout mechanisms, including thermodynamic, as referenced by Reviewer 1, and sea swell, as referenced by Reviewer 2, as a potential breakout mechanism. Regarding themodynamic drivers, we note that the entire water column in McMurdo Sound during winter is characterised by cold and very cold (supercooled) water outflow from underneath the McMurdo Ice Shelf (Leonard et al., 2006), and the water column is nearly isothermal and very close to its freezing point (Lewis and Perkin, 1985; Mahoney et al., 2011). During summer, surface waters flowing into the sound from the northeast exhibit higher temperatures (above the freezing point but mainly below $0\,^{\circ}$C) (Robinson et al., 2014), but this is not the case during the months this study examines. Regarding sea swell, we agree that a study of sea swell effects on McMurdo Sound fast ice would be profitable. However, we note that there was no sea swell data available during the period of our study, as the frequency at which the sea level was recorded by the tide gauge was not sufficient to resolve waves. Thus, the available data in the region and the scope of the manuscript did not allow us to study the effect of sea swell on the McMurdo Sound fast ice in 2019 or the synoptic weather patterns associated with the AWS data. We agree with Reviewer 2 that due to the absence of fetch upstream of the fast ice in the direction of the wind, no significant swell can be formed prior to the polynya having opened. We note that previous studies on fast ice break up in Antarctica did find that the fast ice broke without a clear association to a wave event (Voermans et al., 2020).

We make the following response to the specific comments from Reviewer 2 below. Reviewer comments have been italicized for clarity.

R2 comment:

*The 2019 anomalous breaking of fast ice appears to be associated with KWI and sea ice concentration, as shown in Figure 3. However, this manuscript did not explain the mechanism of fast ice breaking. What mechanism do the authors consider for the fast ice break up by the katabatic wind? Figure 3 showed that the KWI increase coincides with the fast ice break up. When the KWI was large, strong winds were blowing from the south (continent). Again, how does the fast ice is broken by this wind? It is widely known that sea swells affect the breaking of fast ice. The swell effect was also discussed in Banwell et al. (2017), cited by the authors. It seems hard to destroy fast ice only by katabatic wind, even if it is a strong wind. Furthermore, since the wind blows from the shore to the offshore, it is expected not to generate swells that destroy the fast ice.*

Our response:

We agree with Reviewer 2 that due to the absence of fetch upstream of the fast ice in the direction of the wind, no significant swell can be formed prior to the polynya having opened. Previous studies on fast ice break up in Antarctica did find that the fast ice broke without a clear association to a wave event (Voermans et al., 2020), additionally, clear relationships between polynya opening and offshore winds have been found (Ebner et al., 2013; Dale et al., 2017). Without wave data, we cannot speculate on the causes of the fast ice breaking beyond the observation that it is correlated with high strong southerly surface winds, as discussed in the manuscript.

However, we strongly suggest that the drag force due to a strong offshore wind blowing over the surface of the fast ice, particularly early in the season when it is relatively thin, can advect the fast ice to the north, and coupled with secondary effects (such as thermal cracking, etc.), can lead to the break-up of the fast ice cover, e.g. Bogardus et al (2020). Once the fast ice cover thickens and strengthens later in the season, we hypothesise that it becomes strong enough to resist this forcing (see lines 111 – 112 in the manuscript). However, as we do not have data on the strength of the fast ice, we are unable test this hypothesis in this study.

R2 comment:

*Since the katabatic wind is a strong wind from inside the continent, it is expected that the air temperature will drop during the period when the KWI is large. However, as is clear from Figure 3, the temperature rose when the KWI is large. Please explain the reason for this. Is this because of the breaking of fast ice or a coastal polynya formation? Both of them will increase heat flux from the ocean to the atmosphere.*

Our response:

We provided the reasoning for this in the manuscript (lines 75 – 79), which we have stated again here for completeness: "These winds are typically connected to warm temperatures and low air pressure (e.g. Coggins et al., 2013; Chenoli et al., 2013). During winter, higher wind speeds (above $4 – 6$ m s$^{-1}$) increase the mixing between the cold surface inversion layer over the ice shelf and the warmer overlying atmosphere (Cassano et al., 2016). The increase in temperature may also be partly influenced by the Föhn effect, whereby katabatic winds from the Transantarctic Mountains, one of the sources of southerly winds on the Ross Ice Shelf (Parish et al., 2006), experience adiabatic warming." We believe this sufficiently explains why surface air temperatures rise during katabatic wind events.

R2 comment:

*Regarding the fast ice break up during June – July: The reviewer cannot know the details because the authors only show the southerly wind component, but wondering the influence of low pressure rather than the katabatic wind from the following facts: the wind speed increased, the temperature rose, and the atmospheric pressure decreased (Fig. 3). If so, the reviewer considers that fast ice could be collapsed by sea swell. The authors also described it as a "storm event" in their conclusion (P. 4, L. 108). Is this an atmospheric event due to the katabatic wind only? Otherwise, is it the effect of a low-pressure system? Please clarify this.*

Our response:

We addressed the comment on sea swell as a potential fast ice break up mechanism in a previous response. Here

we focus our response on whether a "storm event" is due to katabatic winds only.

The scope of our brief communication manuscript and the available data did not allow for us to attempt to partition the strong southerly surface winds we observed into synoptic and katabatic components. The strong southerly wind events in McMurdo Sound, associated with low pressure and rising air temperatures, are linked to the Ross Ice Shelf Air Stream (RAS), which is forced by synoptic low pressure systems along the coast and fed by katabatic winds near the grounding zone of the Ross Ice Shelf. The RAS is known to occur frequently during winter. A detailed description of this phenomenon would be beyond this brief communication, but we refer Reviewer 2 to Dale (2020) for a discussion for the RAS and its effect on polynyas. We used the term storm event because our KWI index is not based on wind speed and thus we cannot speak of a strong wind event. Specifically, we do not use the wind speed data directly to define the KWI as there are known issues with topographical steering at the Scott Base measurement site, as well as general issues with rime and snow cover interfering with AWS wind measurements in polar regions (Ebner et al., 2013).

Further, strong southerly winds, for which the KWI is used as a proxy, are not in all cases associated with katabatic winds, although that is how they are commonly referred to, which is why we do not speak of katabatic wind events in the manuscript. Katabatic winds play a role in forming the RAS. For a discussion on the link between synoptic winds, katabatic winds and polynya opening, we refer Reviewer 2 to Ebner et al. (2013) who show a clear link between offshore winds (both of synoptic and katabatic origin) and polynya opening. With this said, however, we take onboard the reviewer's concern with our description of the origin of the wind-induced break-up, and acknowledge that the "Katabatic Wind Index" may not be the best name for this index. We will, therefore, rename the index to something along the lines of a "Modified Storm Index" in the revised manuscript to reflect this. This name references directly the origin of the index in Brunt et al. (2006), as well as the modifications we have applied to be able to utilise the index for discrete events. We note that Brunt et al. (2006) found that "... the characteristics of weather conditions most influential in breaking up fast ice... [were] characterized by the simultaneous occurrence of low pressure and an anomalous temperature".

R2 comment:

*This study showed a relationship between a coastal polynya and KWI (section 4). By what mechanism does the polynya cause the fast ice break up? Is it just a description of a relationship between KWI (southern wind) and polynya? The air temperature was below -10 degrees Celsius during the period. Under such atmospheric conditions, even if an open water fraction appears by the divergent ice motion due to prevailing wind, the ocean surface will be immediately covered with thin sea ice. In winter, coastal polynyas should be considered as thin ice-covered areas with high ice concentration rather than low ice concentration areas under the passive microwave sensor's coarse spatial resolution. Many sea ice concentration algorithms used for passive microwave satellite data underestimate the concentration in thin ice-covered areas. It may be possible to regard the low ice concentration region as a coastal polynya signal due to this characteristic, but caution will be required. It does not detect coastal polynyas precisely. For the detection of coastal polynyas from passive microwave satellite data, Tamura et al. (2007; 2008) and Nihashi and Ohshima (2015) would be helpful.*

Our response: We thank Reviewer 2 for highlighting the challenges in using passive microwave satellite data to identify polynya area, and for providing three references that provide information on the development of microwave thin ice algorithms and their application to polynya detection. We will describe this in the revised manuscript and reinforce that coastal polynyas cannot be precisely detected using microwave-derived sea ice concentrations. We also refer Reviewer 2 to Ebner et al. (2013) who used a combination of passive microwave and MODIS data to identify polynyas in their study. They stated that "The two main problems when getting the correct POLA from MODIS data are the spatial coverage of the daily composites and the errors in cloud detection over sea ice in winter. If the main polynya region in the MODIS composite is covered by clouds, an underestimation of POLA is unavoidable, even with the applied correction method. A false detection of clouds can strongly influence the MODIS thin-ice retrieval and, hence, influence the POLA signature. On the other hand, a main problem of AMSR-E data is the relatively coarse horizontal resolution (6.25 km). Small-scale polynyas are not resolvable with this resolution and, hence, the AMSR-E data tends to underestimate the POLA compared to the MODIS thin-ice thickness approach. In addition, thin ice-covered polynyas may be undetected by AMSR-E." (Ebner et al., 2013) (Note that POLA refers to "polynya area".) Hence, we acknowledge that both MODIS and AMSR-E have issues in detecting polynyas, but note that we found good agreement between the two products in this study, as well as with our manual interpretation of SAR imagery, giving us confidence that we can use these products effectively to detect the presence of polynyas.

Regarding the connection between the polynya opening and the fast ice break up, we believe that they are both driven by the same process, i.e. the strong southerly (offshore) surface winds.

**2 References**

Bogardus, R., Maio, C., Mason, O., Buzard, R., Mahoney, A. and de Wit, C.: Mid-Winter Breakout of Landfast Sea Ice and Major Storm Leads to Significant Ice Push Event Along Chuckchi Sea Coastline, Front. Earth Sci., https://doi.org/10.3389/feart.2020.00344, 2020.

Brunt, K. M., Sergienko, O., and MacAyeal, D. R.: Observations of unusual fast-ice conditions in the southwest Ross Sea, Antarctica: preliminary analysis of iceberg and storminess effects, Annals of Glaciology, 44, 183–187, https://doi.org/10.3189/172756406781811754, 2006.

Dale, E.: Interactions Between Changing Weather Patterns and the Antarctic Cryosphere in the Ross Sea Region, PhD Thesis, University of Canterbury, New Zealand, 2020.

Dale, E. R., McDonald, A. J., Coggins, J. H. J., and Rack, W.: Atmospheric forcing of sea ice anomalies in the Ross Sea polynya region, The Cryosphere, 11, 267–280, https://doi.org/10.5194/tc-11-267-2017, 2017.

Ebner, L., Heinemann, G., Haid, V. and Timmermann, R.: Katabatic winds and polynya dynamics at Coats Land, Antarctica, Antarctic Science, 26(3), 309-326, https://doi:10.1017/S0954102013000679, 2013.

Leonard, G H., Purdie, C. R., Langhorne, P. J., Haskell, T. G., Williams, M. M. J., Frew, R. D.: Observations of platelet ice growth and oceanographic conditions during the winter of 2003 in McMurdo Sound, Antarctica, J. Geophys. Res., 111, C04012, doi:10.1029/2005JC002952, 2006.

Lewis, E. L. and Perkin, R. G.: The Winter Oceanography of McMurdo Sound, Antarctica. In Oceanology of the Antarctic Continental Shelf, S.S. Jacobs (Ed.), https://doi.org/10.1029/AR043p0145, 1985.

Mahoney, A. R, Gough, A. J., Langhorne, P. J., Robinson, N. J., Stevens, C. L, Williams, M. M. J., and Haskell, T. G.: The seasonal appearance of ice shelf water in coastal Antactica and its effect on sea ice growth, J. Geophys. Res., 116, C11032, doi:10.1029/2011JC007060, 2011.

Robinson, N. J., Williams, M. J. M., Stevens, C. L., Langhorne, P. J., and Haskell, T. G.: Evolution of a supercooled IceShelf Water plume with an actively growing subice platelet matrix, Journal of Geophysical Research: Oceans, 119, 3425–3446, https://doi.org/10.1002/2013JC009399, 2014.

Voermans, J., Rabault, J., Filchuk, K., Ryzhov, I., Heil, P., Marchenko, A., Collins, C., Dabboor, M., Sutherland, G., and Babanin, A.: Experimental evidence for a universal threshold characterizing wave-induced sea ice break-up, The Cryosphere Discussions, pp. 1–24, https://doi.org/10.5194/tc-2020-201, 2020.

---

## Author Response (AR1)

**Leonard et al. tc-2020-352 author comments**

**20 June 2021**

This document contains the point-by-point reply to the reviewer and editor comments for the Leonard et al. tc-2020-352 manuscript. Reviewer and editor comments are indicated in italics. We provide our responses to Reviewer 1, followed by our responses to Reviewer 2 and finally our responses to the Editor comments.

**1** Responses to Major Comments from Reviewer 1**

We thank Reviewer 1 for their comments on our submission as a Brief Communication within criteria (c): "to disseminate information and data on topical events of significant scientific and/or social interest". We provide the following responses to the points that they have raised.

The authors present a convincing correlation between storm events and fast ice breakout, indicating that it's probably a direct wind-driven (i.e., dynamical) breakout mechanism (and I agree that this is almost certainly the case) - however no alternative mechanisms are discussed. Other studies have indicated that fast ice may be weakened thermodynamically by basal melt (e.g., Arndt et al., 2020, also TC - however this study implied that summertime mode 3 water incursions were important, which is surely not a factor in the winter). I'm not so familiar with the structure of the water column in the Sound during winter, but is it possible that these wind and polynya events enhance vertical mixing - and if warm water (e.g., mCDW) exists on the shelf here, might its entrainment induce basal fast ice melt? And the lag apparent between some storm events and the breakout might also imply a thermodynamic connection (although I accept your explanation involving the land mask of the sea ice concentration data in probably correct). Looking at Pritchard et al 2012 ("Antarctic ice-sheet loss driven by basal melting of ice shelves"), I can see that there's likely no warm water here so my hypothesis is quite unconvincing - but a brief discussion around alternative mechanisms would be appreciated!

To address this comment (as well as a similar comment from Reviewer 2), we have added a Discussion section to the manuscript where in the first paragraph we briefly investigate potential break-out mechanisms (lines 134 - 146 in the revised manuscript). Regarding the possibility of wind and polynya events enhancing vertical mixing, we have added the following sentence:

"Furthermore, the entire water column in McMurdo Sound during winter is conditioned by supercooled water flowing out from the McMurdo Ice Shelf cavity (Leonard et al., 2006), resulting in it being nearly isothermal and very close to its freezing point (Lewis and Perkin, 1985; Mahoney et al., 2011), suggesting that any water upwelled from the opening of the MSP would not cause melting of the fast-ice cover."

**2 **Responses to Minor Comments from Reviewer 1**

6: add "timing of" between "between" and "break-out"

Done.

14: Brett et al ref needs year.

Done.

18: add "stable" between "the" and "fast"

Done.

32: "activity" here is a little ambiguous. You mean sea ice production, right?

We have removed "enhanced MSP activity" and replaced it with "more frequent opening of the MSP".

35: Probably best to avoid starting a sentence with a number (2019).

We have added "The" at the start of this sentence.

41: The Fraser et al 2020 dataset gives 15 day composite maps, not 14 day.

Changed "14" to "15".

47: The "biased" in here implies that these studies didn't correctly account for the icescape change. Is this what you really mean - if so, for both studies?

Changed "biased" to "influenced".

58: Was this IW mode Sentinel-1 imagery? What resolution?. Also Hall and Riggs refs need years.

We have added a sentence (reference lines 62 - 64) giving more information on the Sentinel-1 imagery used. It reads "The SAR imagery used in this study were a combination of Extra Wide (EW) medium resolution imagery (40 m pixel size) and Interferometric Wide (IW) high resolution mode imagery (10 m pixel size resampled to 40 m pixel size for this study)." We have added the years to Hall and Riggs references.

59: "MSP event" is a little ambiguous. Do you mean a large polynya size event? Also here, I'm curious how an active polynya looks in ice surface temperature - presumably a warm temperature? Or is it masked because largely open water?

Yes, we mean a large polynya-size event. We have added the following definition (lines 59 - 60 in the revised manuscript): "(a "polynya event" being defined here as a polynya opening that impacts on the previously established fast-ice cover)." We have added more description to what we observed in the MODIS IST data during a polynya event (reference lines 65 - 74 of the revised manuscript).

65: "Manually identified events" is a little ambiguous. Events of what?

Changed to: "Manually-identified fast-ice break-out events"

74: "connected to" -> "associated with"

Changed "connected to" to "associated with".

75: "warm temperatures" - what temperature? I presume near-surface air temp?

Changed to: "warm near-surface air temperatures".

79: "are correlated to sea ice concentration" - ambiguous description. High SIC? Low SIC? And isn't "correlated with" better than "correlated to"?

Changed to: "correlated with low sea-ice concentration".

82: By "freeze-up" do you mean pack or fast ice?

Changed to: "... the months preceding the formation of a stable fast-ice cover in McMurdo Sound ...".

85: It first struck me as a little unusual to define a KWI without using wind data. What happens if a low pressure system occurs over the central Ross Sea - doesn't this also bring warm air and low pressure? Or is this the effect

you're trying to capture - and these pressure systems enhance the katabatics? A little more clarity here would be appreciated.

As described more below in our response to Reviewer 2, we have renamed the KWI as MSI (Modified Storm Index) to reinforce that the southerly winds in McMurdo Sound result from an interaction of katabatic winds with synoptic-scale low pressure systems, as the reviewer has correctly inferred in their comment.

91: "break-out events" - do you mean fast or pack?

Changed to: "break-out of the fast ice."

117: Although a brief communication, the "big picture" could do with a little more expansion. E.g., this is one of few case studies on fast ice stability, an area where more research is needed, etc. It occurs to me that this region might be a good one for testing forthcoming fast ice tensile strength parameterisations in prognostic fast ice models (e.g., Lemieux et al., 2016, "Improving the simulation of landfast ice by combining tensile strength and a parameterization for grounded ridges"). Also, are there other regions you know of which have a similar fast ice regime (i.e., deep embayment and lack of grounded icebergs) to which the results of this study might be applicable?

We have moved the content of the Conclusions section in the original manuscript to a new Discussion section, and have inserted a new paragraph in the Conclusions (reference lines 164 - 171 in the revised manuscript) where we place this study into the wider context of fast-ice research.

119: The Fraser et al., 2020 dataset is missing from the availability section.

We have added the following statement in the data availability section (reference lines 174 – 176 in the revised manuscript): "Circum-Antarctic landfast sea ice extent data accessed from the Australian Antarctic Data Centre at https://data.aad.gov.au/metadata/records/AAS 4116Fraserfasticecircumantarctic."

*Fig 1: It would be helpful to please annotate the area of active polynya in each SAR image (manually is fine). Similarly for the fast ice edge.*

Active polynya areas in Figure 1 are now shown outlined in orange and magenta lines now show fast-ice edges. These have both been manually picked. We have also added the 77.6° S parallel as a green dashed line and added a label for Cape Royds. We have had to remove the symbols and labels for Mt. Erebus and Mt. Terror to accomplish this.

Fig 2: Does the truncation of the upper half of each wind rose remove any/much information? I'd quite like to see the whole thing (if there's detail in the northerly half) but happy to stick with the half roses if no wind from that half.

The winds in the upper half of each wind rose were not particularly informative, as the northeasterly winds are prevailing and of lower wind speeds, meaning they overwhelmed the wind roses but did not contribute much. However, we have revised Figure 2 to include the full wind roses in order to give the reader the entire picture, and be consistent with Figure 3, which now shows all wind speeds and directions for 2019. We have also added the 90th percentile and above winds for the periods 1997 – 2018 (column three) and 2019 (column four) to further illustrate the extreme nature of the 2019 season. We have replaced the original continuous colour bar with a discrete colour bar in this figure.

**3 Responses to Major Comments from Reviewer 2**

We thank Reviewer 2 for their comments on our submission as a Brief Communication within criteria (c): "to disseminate information and data on topical events of significant scientific and/or social interest". We provide the following responses to the points that they have raised.

The 2019 anomalous breaking of fast ice appears to be associated with KWI and sea ice concentration, as shown in Figure 3. However, this manuscript did not explain the mechanism of fast ice breaking.

The authors used KWI and southerly winds. I straightforwardly regarded these as due to katabatic wind. What

mechanism do the authors consider for the fast ice break up by the katabatic wind? Figure 3 showed that the KWI increase coincides with the fast ice break up. When the KWI was large, strong winds were blowing from the south (continent). Again, how does the fast ice is broken by this wind? It is widely known that sea swells affect the breaking of fast ice. The swell effect was also discussed in Banwell et al. (2017), cited by the authors. It seems hard to destroy fast ice only by katabatic wind, even if it is a strong wind. Furthermore, since the wind blows from the shore to the offshore, it is expected not to generate swells that destroy the fast ice.

We have addressed this comment in the following ways:

(1) We have renamed the Katabatic Wind Index (KWI) to the Modified Storm Index (MSI) and provided a more in-depth description of how katabatic winds and synoptic-low pressure systems interact to produce southerly wind events in McMurdo Sound (reference lines 85 - 90 in the revised manuscript).

(2) We have referenced the work of Dale et al. (2017) and Ebner et al. (2013) to present observed linkages between offshore winds with both katabatic and synoptic components and an increase in polynya area due to wind forcing (reference lines 90 - 96 in the revised manuscript).

(3) We have added a Discussion section where in the first paragraph we briefly investigate potential break-out mechanisms (lines 134 - 146 in the revised manuscript). Regarding the comment of whether sea swell could be a potential break-out mechanism, we have added the following sentence: "Finally, we would not anticipate wave action playing a significant role in breaking up the fast-ice cover due to the absence of upstream fetch associated with the southerly winds." (reference lines 145 - 146 in the revised manuscript).

Since the katabatic wind is a strong wind from inside the continent, it is expected that the air temperature will drop during the period when the KWI is large. However, as is clear from Figure 3, the temperature rose when the KWI is large. Please explain the reason for this. Is this because of the breaking of fast ice or a coastal polynya formation? Both of them will increase heat flux from the ocean to the atmosphere.

We have provided an explanation for how interaction between katabatic winds and synoptic-scale low pressure systems can produce strong southerly surface winds that are warmer than the ambient surface air temperature (reference lines 85 - 90 in the revised manuscript).

Regarding the fast ice break up during June-July: The reviewer cannot know the details because the authors only show the southerly wind component, but wondering the influence of low pressure rather than the katabatic wind from the following facts: the wind speed increased, the temperature rose, and the atmospheric pressure decreased (Fig. 3). If so, the reviewer considers that fast ice could be collapsed by sea swell. The authors also described it as a "storm event" in their conclusion (P. 4, L. 108). Is this an atmospheric event due to the katabatic wind only? Otherwise, is it the effect of a low-pressure system? Please clarify this.

We have revised Figure 2 to include all wind directions in the wind roses and have added two new columns that show the  $90^{th}$  percentile and above winds for 1997 - 2018 and 2019. As mentioned above, we have referenced the work of Dale et al. (2017) and Ebner et al. (2013) to present observed linkages between offshore winds with both katabatic and synoptic components and the increase in polynya area due to wind forcing (reference lines 90 - 96 in the revised manuscript) to support our premise that fast-ice break-out is due to opening of the McMurdo Sound Polynya as a result of wind forcing. We have provided an explanation for how storm events result from interactions between katabatic winds and synoptic-scale low pressure systems (reference lines 85 - 90 in the revised manuscript). We have also interrogated sea-level data from Cape Roberts (5 minute resolution), which showed no indication of anomalous sea level at the times of break-out events.

This study showed a relationship between a coastal polynya and KWI (section 4). By what mechanism does the polynya cause the fast ice break up? Is it just a description of a relationship between KWI (southern wind) and polynya? The air temperature was below -10 degrees Celsius during the period. Under such atmospheric conditions, even if an open water fraction appears by the divergent ice motion due to prevailing wind, the ocean surface will be immediately covered with thin sea ice. In winter, coastal polynyas should be considered as thin ice-covered areas with high ice concentration rather than low ice concentration areas under the passive microwave sensor's coarse spatial resolution. Many sea ice concentration algorithms used for passive microwave satellite data underestimate the concentration in thin ice-covered areas. It may be possible to regard the low ice concentration region as a coastal polynya signal due to this characteristic, but caution will be required. It does not detect coastal polynyas precisely. For the detection of coastal polynyas from passive microwave satellite data, Tamura et al. (2007; 2008) and Nihashi and Ohshima (2015) would be helpful.

We have addressed the comment regarding the break-up mechanism above. We thank Reviewer 2 for highlighting the challenges in using passive microwave satellite data to identify polynya area, and for providing three references that provide information on the development of microwave thin ice algorithms and their application to polynya detection. We have added two sentences in Section 2.1 that acknowledge the challenges in detecting coastal polynyas from passive microwave satellite data and provide reasoning for introducing a sea-ice fraction metric derived from ARTIST sea ice concentrations to characterise regional changes in the sea-ice cover, as opposed to attempting to discern polynya area from passive microwave satellite data (references lines 77 - 80 in the revised manuscript).

**4 Responses to Minor Comments from Reviewer 2**

Reviewer 2 did not provide any minor comments in their review.

**5 Responses to Major Comments from the Editor**

We thank the editor for their comments on our submission as a Brief Communication within criteria (c): "to disseminate information and data on topical events of significant scientific and/or social interest". We provide the following responses to the points that they have raised.

My main concern with your ms and the response to the reviews focusses on differentiating the temperature effects (aka warming) and the forcing from air-pressure and wind-stress changes. I believe that the katabatic you refer to is cool gravity wind, while the Foehn is warm wind. The underlying difference is that the katabatic is a surface wind, while the Foehn wind arises from processes much higher in the atmosphere (Speirs et al., 2010). With this in mind, I invite you to review the definition of the Katabatic Wind Index

We thank the editor for their comment regarding katabatic winds and their invitation to review the definition of the Katabatic Wind Index. As stated above, we have renamed the Katabatic Wind Index (KWI) to the Modified Storm Index (MSI) and provided a more in-depth description of how katabatic winds and synoptic-low pressure systems interact to produce southerly wind events in McMurdo Sound (reference lines 85 - 90 in the revised manuscript). The revised name references directly the origin of the storm index introduced by Brunt et al. (2006), where "modified" refers to adaptations we made to apply the index to discrete events.

The data/images presented do not provide the reader with sufficient information to judge how anomalous the 2019 fast-ice break-out rates in the overall record.

We acknowledge that it was not possible to infer the anomalous wind conditions in 2019 from Figure 2 in the original manuscript. To address this, we have produced a revised Figure 2 that now shows the surface-wind speeds from all directions and the 90th percentile and above surface winds for the years 1997 – 2018 and 2019, respectively. By comparing the 90th percentile wind roses in June and July for the periods 1997 – 2018 and 2019, we believe Figure 2 now clearly shows that the frequency of intense (e.g., 90th percentile and above) southerly winds in June and July 2019 was greater than for the period 1997 – 2018.

We acknowledge that the ARTIST record of sea-ice concentrations is relatively short (2013 - 2019) as compared to the data records for wind speed, surface air temperature and MSLP. We chose not to use more spatially-coarse remotely-sensed sea-ice concentration data due to the fact that the larger footprints do not sufficiently resolve the sea-ice conditions in McMurdo Sound. This has meant that we have a shorter data record to assess winter 2019 break-out events against (Figure 3a) as we do for MSI identified storm events (Figure 3b).

Accepting that some years are influenced by the presence of a large iceberg, the authors should provide a longer record, i.e., to earlier years.

We agree that the number of iceberg-affected years in the data record has affected our ability to construct a longer time record of fast-ice conditions in McMurdo Sound. The two remote-sensing derived sea ice products we have used in this study unfortunately do not extend farther back than 2000 for the Fraser et al. fast ice extent dataset and 2012 for the Spreen et al. (2008) dataset, and the meteorological dataset is only available back to 1996. As our

goal with this brief communication is to report on a topical event of scientific interest, we felt that investigating ways to extend these records farther back in time was beyond the scope of the study.

**How do the thermodynamic and the dynamic components of the KWI relate to each other? And which one is crucial in driving the ice breakout? Is this consistent over the full data record?**

As mentioned above, we have provided an explanation for how interaction between katabatic winds and synopticscale low pressure systems can produce strong southerly surface winds that are warmer than the ambient surface air temperature (reference lines 85 - 90 in the revised manuscript). Also as mentioned above, we have added a Discussion section where in the first paragraph we briefly investigate potential break-out mechanisms (lines 134 - 146 in the revised manuscript), and we have referenced the work of Dale et al. (2017) and Ebner et al. (2013) to present observed linkages between offshore winds with both katabatic and synoptic components and the increase in polynya area due to wind forcing (reference lines 90 - 96 in the revised manuscript). On the balance of this, we put forth that it is wind-forcing that is the crucial driver of ice break-out, but we are unable to quantify other effects such as sea-ice internal temperature, sea swell and ocean current / temperature.

What are the elevations for the temp and pressure sensors at Scott Base? And over which surface type are deployed? Glacial ice, rock, ...

We have added that the elevation of the wind, pressure and temperature sensors at Scott Base is 20 m (reference line 106 in the revised manuscript). The sensors are located in the immediate vicinity of Scott Base and the surface type is rock.

One would expect a clearer katabatic signal of the KWI would be a function of the pressure gradient, and even more if also considering a temperature gradient (normalized relative to freezing temperature).

We have renamed the KWI as MSI (Modified Storm Index) to reinforce that the southerly winds in McMurdo Sound result from an interaction of katabatic winds with synoptic-scale low pressure systems.

As KWI is the product of Tair times Pair, how do the authors relate this to a mechanism that leads to the breakup of fast ice in McMurdo Sound?

The MSI (renamed from KWI in the original manuscript) is constructed from the product of normalised positive temperature and negative pressure anomalies, following what was developed by Brunt et al. (2006) to represent conditions of "storminess" that most consistently led to abrupt break-up, mobilization and clear-out of fast ice in the southwest Ross Sea. Following this definition, and as stated above, we then relate the MSI to the opening of the McMurdo Sound Polynya, which results in the breaking-up of regions of the fast-ice cover.

Also, was any possibly interference with tidal effects considered? If peak tidal elevation would have coincided with the onset of the MSLP gradient, then a plausible mechanism for the fast-ice breakout would have been identified.

We had investigated this, as noted above, and found no evidence of tidal-induced break-out. We have addressed this comment by adding the following text (reference line 141 in the revised manuscript) "..., and we did not observe any correlation between tidal state and the 2019 break-out events in this study."

**6 Responses to Minor Comments from the Editor**

Decide on hyphenation rule and apply consistently throughout the manuscript. For example "fast-ice cover" (line 113) versus "fast ice cover" (line 111). Pls apply this rule to all three (or more) noun sequences. Typical recommendation is to hyphenate the first two nouns to establish the context.

We have made these changes throughout the entire revised manuscript.

Why introduce terminology such as "sea ice fractional cover" and "sea-ice fraction" when there is a standard, namely "sea-ice concentration"? Suggest to replace with the latter throughout the manuscript.

We have provided a reason for this approach as (reference lines 123 - 125 in the revised manuscript): "This method was chosen in preference to computing an average sea ice concentration as it is better suited to quantifying changes

in sea ice extent, which we equate generally to changes in the fast-ice cover in the sound."

The "o N/E/S/W" is used inconsistently... with regard to the spacings. Please climatological mean"?

We have standardised the spacing for  $^{\circ}$  N/E/S/W throughout the revised manuscript. The term "climatological mean" is applied to surface air temperatures and MSLP and is the mean value of the respective quantities over the time period 1997 – 2018, smoothed by a 3-day window.

What are the elevations for the temp and pressure sensors at Scott Base? And over rectify.

The elevations are 20 m. This information has been added in line 106 of the revised manuscript.

Correct "e.g." to "e.g.," throughout the manuscript.

Done.

51: Change "Sound" to lower case.

Done.

52: Change "Sound" to lower case.

Done.

58: Add publication year to "Hall and Riggs, a, b)".

Done.

67: Change "Sound" to lower case.

Done.

86: Definition of the KWI: Clarify how the hourly air temperature and mean sea level pressure data/anomalies are used relative to the "climatological mean". I.e., what is the frequency of the "climatological mean"

The climatological mean is calculated from hourly data that has been smoothed by a running 3-day window for the period 1997 - 2018 for temperature, and 2002 - 2018 for MSLP, respectively. The anomalies are the differences between the hourly 2019 data and the hourly 3-day smoothed climatological means.

118: Add publication year to "Brett et al."

Done.

119: Correct "data is" to "data are". TWICE in this line.

Done for the first one. The second one was removed as we no longer include Linda AWS data in the revised manuscript.

123: No need to provide detail on which exact data sets were processed/analyzed by whom. Shorten to "GHL had the initial idea for this study and analysed the satellite imagery/products, KET".

Done.

Fig. 1:

Add "the" to read "inset shows the location".

Replace "from" with "for" to read "for which the sea ice fractions".

b), c) and d): Define acronyms shown: "MSP" and "RSP".

b) - f): Add year to dating provided.

All done.

Fig. 2: Need to add information to place the single winter 2019 into the winter mean for 1998 - 2018. Either add standard deviations are add data/images for all other individual winters. – Lack of this information presents a crucial omission and does not allow the reader to assess the severity of the atmospheric forcing on the fast ice.

We thank the editor for this very important comment, and we have addressed this by adding the  $90^{\text{th}}$  percentile and above winds for the periods 1997 - 2018 (column three) and 2019 (column four) so that the reader can now assess the severity of the 2019 atmospheric forcing on the fast ice.

*Fig. 3: \* b)* Do the Scott Base data indicate a critical threshold for ice breakout? \* c) Data from Linda and Scott Base are clearly shown in this sub figure.

We do not believe that a critical threshold for ice break-out can be inferred from the Scott Base atmospheric conditions alone as the thickness / strength of the fast-ice cover makes too significant of a contribution.

We have removed Linda AWS data from Figure 3c, and replaced the combined Scott Base and Linda wind speeds (original panel c) with Scott Base wind speeds (new panel c) and directions (new panel d). We have used red dots to identify the wind speeds and directions for  $90^{th}$  percentile and above wind events. In the new panel c, we have also denoted the monthly  $90^{th}$  percentile wind speeds for the period 1997–2018 by a red line.

---

## Author Response (AR2)

**Authors' Response to Editor and Reviewer 1 Comments of Revised Manuscript**

**September 15, 2021**

This document contains the point-by-point reply to the Reviewer 1 and Editor comments for the revised Leonard et al. tc-2020-352 manuscript. Reviewer and Editor comments are indicated in bold. We provide our responses to Reviewer 1 first, followed by our responses to the Editor.

**1 Reviewer 1 comments**

I consider the manuscript improved overall to be easier to follow than the previous one, thanks to the author's sincere efforts. However, I have two very minor comments, as described below.

**1.1** Minor comments**

P. 1, L. 4 (abstract): "We analyse the 2019 sea-ice conditions and relate them to southerly wind events using a Modified Storm Index (MSI)." I guess MSI includes not only wind but air temperature.

Yes, MSI uses air temperature and mean sea-level pressure. We acknowledge the ambiguity identified by the reviewer and have edited the abstract to read:

"We analyse the 2019 sea-ice conditions and relate them to a Modified Storm Index (MSI), a proxy for southerly wind events".

**P. 2, L43: Icebergs C-19 and B-15A are pretty famous. Even so, timing information when these icebergs affect sea ice in the analysis area is helpful for readers.**

We have added the approximate timing of when these two icebergs were located in and around the mouth of the sound (refer to lines 43–45 in the revised manuscript). For B-15A, this was from approximately January 2001 to November 2004 (Brunt et al., 2006). For C-19, this was approximately September 2002 to May 2003 (Arrigo and van Dijken, 2003). The second reference was not previously included in our reference list; its citation is:

Arrigo, K. R. and van Dijken, G. L.: Impact of iceberg C-19 on Ross Sea primary production, Geophysical Research Letters, 30, https://doi.org/https://doi.org/10.1029/2003GL017721, 2003

The new text is:

This data set covers March 2000 to February 2018 and includes the time when the sea ice in McMurdo Sound was strongly impacted by the presence of very large, tabular icebergs in the mouth of the sound (e.g., B-15A from approximately January 2001 to November 2004 (Brunt et al., 2006) and C-19 from approximately September 2002 to May 2003 (Arrigo and van Dijken, 2003)).

**2 Editor comments**

**For future reviews, it would be great if you as author(s) would kindly supply a response to the comments of the reviewers and the editor. This would speed up the review of subsequent ms versions, but replying in dot points to each comment raised by reviewers and editor.**

We thank the editor for this comment. Although we did go through the editor's comments thoroughly and make changes to the text of the revised manuscript where we identified clarifications were needed, we did not directly address the editor's comments from the first review in our response to reviewer comments because we were aware that these would be posted to The Cryosphere Discussions (TCD). As the editor's comments were not posted to TCD, it was unclear to us if we were meant to provide our responses to the editor's comments in our response to reviewers, therefore, effectively posting the editor's comments to TCD.

We have provided our point-by-point responses to the editor's comments below.

**General comments**

**What are the elevations for the temp and pressure sensors at Scott Base? And over which surface type are deployed? Glacial ice, rock, ... ?**

We have added this information to the revised manuscript (refer to lines 108 - 111). We include this information here as well for completeness:

The Scott Base weather station is located immediately adjacent to Scott Base at  $77.85^{\circ}$  S,  $166.76^{\circ}$  E, approximately 150 m from the coast and on the order of 30 km east of the centre of the sound. The elevation of the station is 20 m AMSL with the sensors deployed at a height of 6–7 m above the ground. The ground surface is is seasonally snow-free scoria and volcanic rock.

**One would expect a clearer katabatic signal if the KWI would be a function of the pressure gradient, and even more if also considering a temperature gradient (normalized relative to freezing temperature).**

We thank the editor for raising this point. We had investigated using the temporal temperature and pressure gradients instead of the anomaly; however, this approach results in a very unclear signal due to the noise of the signal as well as the fact that the temperature increase often precedes the pressure decrease.

**As KWI is the product of Tair times Pair, how do the authors relate this to a mechanism that leads to the breakup of fast ice in McMurdo Sound? – See below query about TD vs Dyn contribution.**

As stated above, we use the KWI (now MSI) as a proxy for southerly wind events, with the wind stress imposed on the sea ice being the primary driver for breakup. We do not propose that either the temperature or the pressure are direct drivers for fast-ice breakup during winter.

**Also, was any possibly interference with tidal effects considered? If peak tidal elevation would have coincided with the onset of the MSLP gradient, then a plausible mechanism for the fast-ice breakout would have been identified.**

We examined the data record from the Cape Roberts tide gauge, located in the northwest of the sound at  $77.0^{\circ}$  S,  $163.2^{\circ}$  E, but the temporal resolution of the data (5 minutes) was not sufficient to link break-out events to ocean swell. We also determined that there were no correlations between break-out events and phase of the tidal cycle.

We have amended the following sentence in the manuscript, adding the details of the data that were interrogated (refer to lines 146 - 147):

Kim et al. (2018) did not find a significant relationship between sea-ice temperature and break-out in their

McMurdo Sound study, and we did not find any correlation between tidal state and the 2019 break-out events in this study, based on our analysis of 5-minute sea-level height data from the Land Information New Zealand tide gauge at Cape Roberts (77.0  $^{\circ}$  S, 163.2  $^{\circ}$  E) in the northwestern sound.

We have also added the data source in the data availability section:

Cape Roberts tide gauge data can be accessed from Land Information New Zealand at https://www.linz.govt.nz/sea/tides/sea-level-data/sea-level-data-downloads

**Specific comments**

We thank the editor for their specific comments, all of which we have implemented. We provide further comment here on a handful of the specific comments that we felt warranted a specific response.

**148–150 Pls add how the pack ice found seaward of the fast ice impacts on holding in the fast ice, i.e. compare the seasonal evolution of the pack ice to that of the fast ice.**

From our observations, the pack ice immediately to the north of the fast-ice edge does provide protection against swell induced breakup, however, in the southerly wind events that we analysed, the pack ice is advected away from the fast-ice edge before the fast-ice breakup initiates (as evidenced by the opening of the McMurdo Sound Polynya); hence the pack ice does not tend to hold the fast ice in place. As we are investigating individual events, we did not analyse the seasonal evolution of the pack ice as compared to the fast ice. Kim et al. (2018) found a correlation between fast-ice extent and total sea-ice concentration (fast + pack ice) in the sound and also noted that fast ice in McMurdo Sound does not seem to follow pack-ice trends in the Ross Sea, though this is probably heavily influenced by the iceberg effect on the fast ice obscuring other signals. They also noted that strong southerlies pushed the pack ice away from the fast-ice edge (in cases where the fast ice did not break out) which would suggest that pack ice offers little to no protection to the fast-ice cover against southerly winds.

**167: Strongly suggest avoid reference to specific sea-ice models such as "CICE" and "LIM". If choosing to retain these, please spell out the model names and add reference.**

We have removed reference to specific models.

172: Is the preferred / best access for ESA's S1 SAR data really via the Alaska SAR Facility? Would think to check at Copernicus data server first.

We routinely use the Alaska SAR Facility to search for and acquire SAR imagery as we find its interface intuitive to use. We feel this is appropriate, even when we are accessing ESA datasets, as per the following statement on the Alaska Sar Facility webpage:

"NASA's provision of the complete ESA Sentinel-1 synthetic aperture radar (SAR) data archive through the ASF DAAC is by agreement between the U.S. State Department and the European Commission (EC). As part of the Earth-observation Copernicus program, the Sentinel mission will provide scientists with accurate, timely, and easily accessible information to help shape the future of our planet. Content on ASF's Sentinel web pages is adapted from the ESA Sentinel-1 website."

We appreciate, however, that we have not adequately acknowledged the European Space Agency as the source of the Sentinel-1 SAR imagery. We have now addressed this by adding the following to the data availability section:

Copernicus Sentinel-1 data, retrieved from the Alaska SAR Facility DAAC at https://search.asf. alaska.edu/ were processed by the European Space Agency.

Caption of Fig 2: Add the bin size for the wind roses. For clarification, add info on wind direction being defined as the direction the wind is originating from.

These additions have been made in the figure caption:

The maximum frequency shown is 3 %, with each circle representing an increment of 0.6 %, and the wind direction defined as the bearing from which the wind originated. The bin size of each spoke is  $10^{\circ}$ .

Finally, in response to a suggestion made by the editor, we have modified some of the text in the Discussion and Conclusions to highlight some of the novel ideas and new information from the study.